# Replication Study: Transcriptional amplification in tumor cells with elevated c-Myc

L Michelle Lewis[1], Meredith C Edwards[1], Zachary R Meyers[1],
C Conover Talbot Jr[2], Haiping Hao[2], David Blum[1],
Reproducibility Project: Cancer Biology*

[1]University of Georgia, Bioexpression and Fermentation Facility, Georgia, United States; [2]Johns Hopkins University, Deep Sequencing and Microarray Core Facility, Maryland, United States

*For correspondence:
tim@cos.io;
nicole@scienceexchange.com

Group author details:
Reproducibility Project: Cancer Biology See page 12

**Abstract** As part of the Reproducibility Project: Cancer Biology, we published a Registered Report (Blum et al., 2015), that described how we intended to replicate selected experiments from the paper 'Transcriptional amplification in tumor cells with elevated c-Myc' (Lin et al., 2012). Here we report the results. We found overexpression of c-Myc increased total levels of RNA in P493-6 Burkitt's lymphoma cells; however, while the effect was in the same direction as the original study (Figure 3E; Lin et al., 2012), statistical significance and the size of the effect varied between the original study and the two different lots of serum tested in this replication. Digital gene expression analysis for a set of genes was also performed on P493-6 cells before and after c-Myc overexpression. Transcripts from genes that were active before c-Myc induction increased in expression following c-Myc overexpression, similar to the original study (Figure 3F; Lin et al., 2012). Transcripts from genes that were silent before c-Myc induction also increased in expression following c-Myc overexpression, while the original study concluded elevated c-Myc had no effect on silent genes (Figure 3F; Lin et al., 2012). Treating the data as paired, we found a statistically significant increase in gene expression for both active and silent genes upon c-Myc induction, with the change in gene expression greater for active genes compared to silent genes. Finally, we report meta-analyses for each result.
DOI: https://doi.org/10.7554/eLife.30274.001

## Introduction

The Reproducibility Project: Cancer Biology (RP:CB) is a collaboration between the Center for Open Science and Science Exchange that seeks to address concerns about reproducibility in scientific research by conducting replications of selected experiments from a number of high-profile papers in the field of cancer biology (Errington et al., 2014). For each of these papers a Registered Report detailing the proposed experimental designs and protocols for the replications was peer reviewed and published prior to data collection. The present paper is a Replication Study that reports the results of the replication experiments detailed in the Registered Report (Blum et al., 2015) for a 2012 paper by Lin et al., and uses a number of approaches to compare the outcomes of the original experiments and the replications.

In 2012, Lin et al. reported results that the c-Myc transcription factor, a potent oncogene that is frequently overexpressed in a large percentage of cancers, globally amplifies the expression of actively transcribed genes, opposed to regulating specific target genes. Using the P493-6 cell line, a model for *MYC* activation in Burkitt's lymphoma, total levels of RNA per cell were reported to increase when c-Myc was highly expressed compared to conditions where c-Myc expression was

low. Additionally, active genes in cells with low c-Myc levels were reported to increase in expression upon c-Myc induction, in contrast to genes that were silent under low c-Myc conditions that did not change.

The Registered Report for the 2012 paper by Lin et al. described the experiments to be replicated (Figure 1B and 3E–F), and summarized the current evidence for these findings (*Blum et al., 2015*). Since that publication there have been additional studies investigating the ability c-Myc to influence the global gene expression output of cells. Similar to Lin et al. other studies have reported c-Myc dependent amplification of cellular RNA (*Hart et al., 2014*; *Hsu et al., 2015*; *Nie et al., 2012*; *Sabò et al., 2014*), although this observation was not reported in all biological systems (*Fagnocchi et al., 2016*; *Sabò et al., 2014*; *Walz et al., 2014*). It has been suggested c-Myc regulates specific genes that indirectly lead to RNA amplification (*Sabò et al., 2014*; *Sabò and Amati, 2014*; *Walz et al., 2014*). This has also been suggested of MYCN (*Duffy et al., 2015*). The reported differences could be a result of the intrinsic variation between cell lines in maintaining the transcriptome (*Trakhtenberg et al., 2016*). Indeed, a recent study reported that distinct transcriptional regulation can be accounted for by differences in promoter affinity under different c-Myc expression levels (*Lorenzin et al., 2016*).

The outcome measures reported in this Replication Study will be aggregated with those from the other Replication Studies to create a dataset that will be examined to provide evidence about reproducibility of cancer biology research, and to identify factors that influence reproducibility more generally.

## Results and discussion

### Conditional expression of c-Myc in the B-cell line P493-6

To test the effects of increased levels of c-Myc on gene expression we used the same human P493-6 B cell line of Burkitt's lymphoma that contains a conditional tetracycline-repressive *MYC* transgene (*Pajic et al., 2000*; *Schuhmacher et al., 1999*) as the original study. We performed Western blot analysis to confirm c-Myc expression could be reduced to very low levels and then reactivated after removal of tetracycline. This is comparable to what was reported in Figure 1B of *Lin et al. (2012)* and described in Protocol 1 in the Registered Report (*Blum et al., 2015*). Since proliferation of P493-6 cells depend on c-Myc expression and the presence of serum (*Pajic et al., 2000*; *Schuhmacher et al., 1999*), with serum reported to stimulate a majority of genes independent of c-Myc (*Schlosser et al., 2005*), we maintained these cells in separate lots of serum to assess whether the results differed. For cells maintained in both lots of serum, treatment with tetracycline resulted in a strong decrease in c-Myc protein levels (*Figure 1A*). After removal of tetracycline, c-Myc levels increased over time approaching the levels observed in tetracycline-free conditions.

### Total RNA levels following c-Myc overexpression

We sought to independently replicate whether increased levels of c-Myc resulted in increased absolute levels of RNA. This experiment is similar to what was reported in Figure 3E of *Lin et al. (2012)* and used the same extraction method for total RNA quantification, which was described in Protocol 2 in the Registered Report (*Blum et al., 2015*). Total RNA was isolated from P493-6 cells 0, 1, and 24 hr after tetracycline release and the amount of RNA per 1,000 cells was quantified (*Figure 1B*). We found that under conditions where c-Myc expression was low (0 hr), there was a mean of 1.55 ng total RNA per 1,000 cells (ng/1 k cells) [n = 3, $SD$ = 0.20], which increased to 1.77 ng/1 k cells [n = 3, $SD$ = 0.31] when c-Myc expression was high (24 hr), a 1.14 times increase, for serum lot one, which was not statistically significant ($t(6)$ = 1.02, $p$=0.347). Serum lot two changed from a mean of 1.57 ng/1 k cells [n = 3, $SD$ = 0.21] at 0 hr to 2.25 ng/1 k cells [n = 3, $SD$ = 0.19] at 24 hr, a 1.43 times increase, which was statistically significant ($t(6$ = 5.03, $p$=0.0024). This compares to the original study, which reported a mean of 4.25 ng/1 k cells at 0 hr, which increased to 5.47 ng/1 k cells at 24 hr, a 1.29 times increase in total RNA levels. In both studies there was a minor decrease at 1 hr after tetracycline release when c-Myc levels begin to become detectable. Total RNA per 1,000 cells at 0 hr were much lower in this replication attempt than those reported in the original study, although

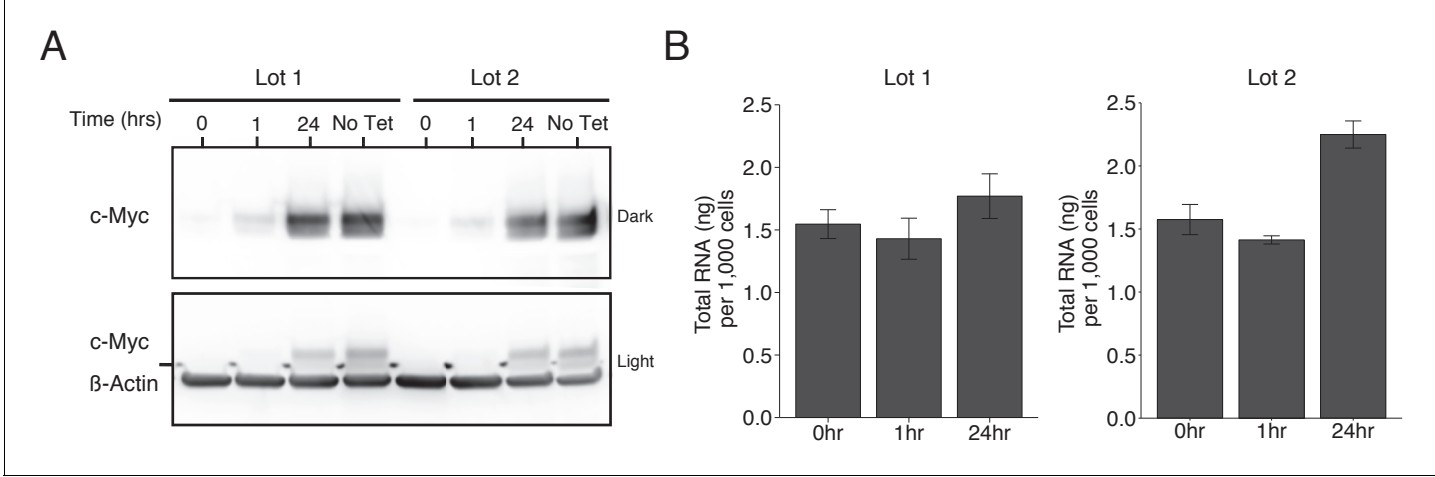

**Figure 1.** Induction of c-Myc in P493-6 cells and impact on total RNA levels. P493-6 cells were grown in the presence of tetracycline (Tet) for 72 hr and switched into Tet-free growth medium to induce c-Myc expression. Cells were cultured in two separate lots of serum. (**A**) Representative Western blot using an anti-c-Myc antibody (top panels) or an anti-ß-Actin antibody (bottom panel). Two exposures of the anti-c-Myc antibody are presented to facilitate detection of c-Myc. (**B**) Quantification of total RNA levels (ng of total RNA per 1,000 cells) for cells at 0, 1, and 24 hr after release from Tet. Means reported and error bars represent s.e.m. from three independent biological repeats. For serum lot one, one-way ANOVA on total RNA levels of all groups; $F_{(2, 6)}=1.25$, $p=0.353$. Planned contrast between 0 hr and 24 hr; $t_{(6)} = 1.02$, $p=0.347$ with *a priori* alpha level = 0.05. For serum lot two, one-way ANOVA on total RNA levels of all groups; $F_{(2, 6)}=21.87$, $p=0.00176$. Planned contrast between 0 hr and 24 hr; $t_{(6)} = 5.03$, $p=0.0024$ with *a priori* alpha level = 0.05. Additional details for this experiment can be found at https://osf.io/tfd57/.
DOI: https://doi.org/10.7554/eLife.30274.002

changes in total RNA levels were in the same direction following c-Myc expression. Similarly, another independent study that measured total RNA from P493-6 cells reported a different level at 0 hr (~3 ng/1 k cells), while also reporting increased levels following c-Myc expression (*Sabò et al., 2014*). There are multiple possible explanations for these differences, such as variation in RNA expression during cell culture passage (*Hiorns et al., 2004*), low yield of the RNA isolation procedure (e.g. incomplete homogenization), or the high variance associated with manual cell counts using a hemacytometer (*Biggs and Macmillan, 1948*; *Nielson et al., 1991*). To summarize, for this experiment we found results that were in the same direction as the original study and not statistically significant for serum lot one, while statistically significant for serum lot two.

## Digital gene expression following c-Myc overexpression

To test whether c-Myc expression amplifies the existing gene expression program, digital gene expression analysis using the NanoString nCounter platform was performed on a set of genes from multiple functional categories. This experiment is similar to what was reported in Figure 3F and Table S1 of *Lin et al. (2012)* and described in Protocols 3–4 in the Registered Report (*Blum et al., 2015*). We quantified mRNA levels/cell of 1369 genes, of which 1212 were the same genes as the 1338 genes interrogated in the original study. We used the same criteria as the original study to classify a gene as silent (expression was less than 0.5 transcript/cell at time 0 hr) or active (more than one transcript/cell at time 0 hr). In cells with low levels of c-Myc (0 hr) there were 708 active genes with a median expression of 4.70, and 580 silent genes with a median expression of 0.032, for serum lot one. For active genes, 75% of the genes increased from 0 hr to 1 hr, 68% increased from 0 hr to 24 hr, and 59% increased from 1 hr to 24 hr upon c-Myc induction. This corresponds to a 1.11, 1.50, and 1.36 times increase in median expression, respectively (*Figure 2*, *Figure 2—figure supplement 1*). For silent genes, 74% of the genes increased from 0 hr to 1 hr, 66% increased from 0 hr to 24 hr, and 50% increased from 1 hr to 24 hr, corresponding to a 1.19 and 1.13 times increase, and a 0.05 times decrease in median expression, respectively (*Figure 2*, *Figure 2—figure supplement 1*). Serum lot two gave similar results, although there were variations in the number of genes identified as silent or active as well as the degree of increase among the conditions (*Figure 2*, *Figure 2—figure supplement 1*). This compares to the original study that identified 755 active genes with a

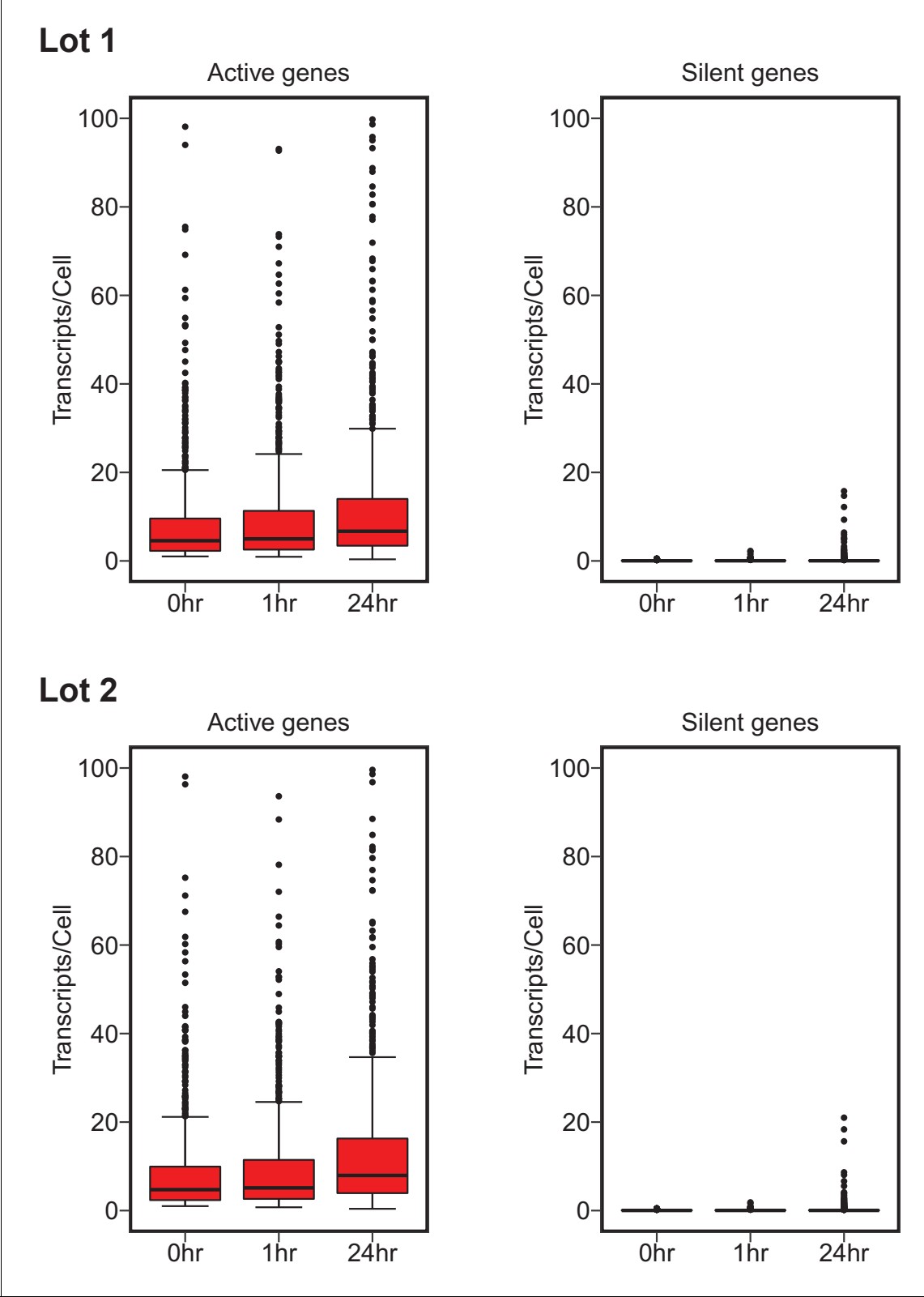

**Figure 2.** Digital gene expression analysis. P493-6 cells grown in the presence of tetracycline (Tet) for 72 hr for repression of the conditional p*myc*-tet construct, were switched into Tet-free growth medium to induce c-Myc expression. Cells were cultured in two separate lots of serum. Transcripts/cell estimates from NanoString nCounter gene expression assays (1369 genes assay) for active (left) and silent (right) genes at 0, 1, and 24 hr after release from Tet. Active genes expressed greater than one transcript/cell. Silent genes expressed less than 0.5 transcript/cell. Box and whisker plots with

*Figure 2 continued on next page*

*Figure 2 continued*

median represented as the line through the box and whiskers representing values within 1.5 IQR of the first and third quartile. Cells grown in serum lot one: active genes = 708, silent genes = 580. Cells grown in serum lot two: active genes = 719, silent genes = 573. Confirmatory analysis is reported in *Table 1* and exploratory statistical analysis is reported in *Table 2* and *Table 3*. Additional details for this experiment can be found at https://osf.io/fn2y4/.

DOI: https://doi.org/10.7554/eLife.30274.003

The following source data and figure supplements are available for figure 2:

**Source data 1.** List of Reporter CodeSets and gene expression values.

DOI: https://doi.org/10.7554/eLife.30274.006

**Figure supplement 1.** Logarithmic expression of genes.

DOI: https://doi.org/10.7554/eLife.30274.004

**Figure supplement 2.** Comparison of gene expression data as continuous.

DOI: https://doi.org/10.7554/eLife.30274.005

median expression of 7.06, and 514 silent genes with a median expression of 0.00 (more than half the silent genes did not have a reported expression value). Active genes in the original study, increased 91% from 0 hr to 1 hr, 96% from 0 hr to 24 hr, and 87% from 1 hr to 24 hr upon c-Myc induction, corresponding to a 1.23, 2.45, and 1.99 times increase in median expression, respectively. Silent genes in the original study, increased 23% from 0 hr to 1 hr, 29% from 0 hr to 24 hr, and 30% from 1 hr to 24 hr, with the median expression unchanged among conditions. In addition, we further examined the extent of overlap of active and silent genes between the original study and this replication attempt. Of the 1212 genes that were interrogated in both studies, 88.8% (603/679) of the active genes we identified in serum lot one were also active in the original study (90.1% (612/679) for serum lot two). For silent genes, 96.4% (456/473) of the genes we identified as silent in serum lot one were common with the silent genes identified in the original study (95.8% (453/473) for serum lot two).

To test whether active genes, as well as silent genes, increased expression during c-Myc induction we performed the confirmatory analysis as outlined in the Registered Report (*Blum et al., 2015*). This analysis differed from what was reported in the original study by analyzing the data as paired instead of unpaired. As suggested during peer review of the Registered Report, this is because expression of the same gene, analyzed across different conditions, is not independent (*Blum et al., 2015*). We performed a Wilcoxon signed-rank test on active genes comparing expression at 0 hr to 1 hr, 0 hr to 24 hr, and 1 hr to 24 hr, which were statistically significant for cells grown in both lots of serum (*Table 1*). The same comparisons were performed on silent genes, which were also statistically significant, with the exception of the silent gene comparison of 1 hr to 24 hr for serum lot one. Considering this was not the test reported in the original study, we conducted these paired analyses on the original data to provide a direct comparison. For both active and silent genes c-Myc induction resulted in statistically significant increases in expression, with the exception of the silent gene comparison from 0 hr to 1 hr (*Table 1*). This is in contrast to the results of the unpaired tests that were reported in the original study where active genes were reported to have a statistically significant increase in expression and silent genes were reported as not statistically significant for all comparisons. We conducted an exploratory unpaired analysis on the replication data for comparison, which resulted in statistically significant differences among the active gene comparisons as well as half of the silent gene comparisons (*Table 2*).

Importantly, though, the question of whether the change in expression among active genes is different than silent genes has not been tested. This would require a separate test on their difference (*Gelman and Stern, 2006*; *Nieuwenhuis et al., 2011*). To test whether active genes increased in expression during c-Myc induction more than silent genes, we performed an exploratory analysis on the difference in expression of active genes during c-Myc induction (e.g. from 0 hr to 24 hr) compared to the difference in expression of silent genes over that same period (e.g. from 0 hr to 24 hr). For both the original and replication data, there was a statistically significant increase in expression of active genes compared to silent genes (*Table 3*). This suggests that active genes and silent genes do not have similar rates of expression upon c-Myc induction. To summarize, for this experiment we found results that were in the same direction as the original study and suggest that while both active and silent genes increased in expression upon c-Myc induction, the rate of increase was different.

**Table 1.** Confirmatory statistical tests.

| Genes | Comparison | Study | Z value | P value | Sample size (n) |
|---|---|---|---|---|---|
| Active | 0 hr vs 1 hr | RP:CB Lot 1 | 14.86 | 1.36e-55 | 708 |
| | | RP:CB Lot 2 | 11.83 | 2.83e-34 | 719 |
| | | *Lin et al. (2012)* | 21.17 | 1.05e-130 | 755 |
| | 0 hr vs 24 hr | RP:CB Lot 1 | 9.922 | 3.67e-24 | 708 |
| | | RP:CB Lot 2 | 12.77 | 3.77e-40 | 719 |
| | | *Lin et al. (2012)* | 23.26 | 3.33e-184 | 755 |
| | 1 hr vs 24 hr | RP:CB Lot 1 | 4.742 | 1.92e-06 | 708 |
| | | RP:CB Lot 2 | 10.04 | 9.91e-25 | 719 |
| | | *Lin et al. (2012)* | 23.26 | 3.33e-184 | 755 |
| Silent | 0 hr vs 1 hr | RP:CB Lot 1 | 12.61 | 7.11e-40 | 580 |
| | | RP:CB Lot 2 | 7.05 | 9.29e-13 | 572 |
| | | *Lin et al. (2012)* | −1.998 | 0.0457 | 274 |
| | 0 hr vs 24 hr | RP:CB Lot 1 | 8.328 | 2.22e-17 | 579 |
| | | RP:CB Lot 2 | 8.156 | 1.03e-16 | 572 |
| | | *Lin et al. (2012)* | 3.179 | 0.00144 | 276 |
| | 1 hr vs 24 hr | RP:CB Lot 1 | 0.6853 | 0.493 | 580 |
| | | RP:CB Lot 2 | 4.436 | 8.35e-06 | 573 |
| | | *Lin et al. (2012)* | 5.806 | 4.11e-09 | 236 |

These confirmatory statistical tests relate to the data presented in **Figure 2**. Wilcoxon signed-rank test, which treat the data as paired, were conducted for the original study (**Lin et al., 2012**) and this replication attempt (RP:CB). Uncorrected *p* values are reported with an *a priori* significance threshold of. 0167. Sample sizes reported are based on the sample size used in the tests. Additional details for this experiment can be found at https://osf.io/fn2y4/.
DOI: https://doi.org/10.7554/eLife.30274.007

The original study and this replication attempt used the same criteria to characterize a gene as silent or active, but there are many negative consequences of dichotomizing continuous variables, such as information loss, especially with a small gene set (**Altman and Royston, 2006**; **Cohen, 1983**). Papers published after the original study took an unbiased view by collecting comprehensive RNA-sequencing data to assess if the transcriptional effects of c-Myc were direct or indirect, concluding c-Myc activates and represses transcription of discrete gene sets, which in turn leads to induced RNA amplification (**Sabò et al., 2014**; **Walz et al., 2014**). Furthermore, Sabò and colleagues also used NanoString technology to quantify a subset of the differentially expressed genes identified by RNA-seq and observed similar results that revealed upward shifts in gene expression upon c-Myc induction (**Sabò et al., 2014**). However, instead of dichotomizing genes as active or silent, gene expression data was presented as continuous. Similarly, we presented the digital gene expression data generated during this replication attempt as continuous, which illustrates a general pattern of overall increased gene expression following c-Myc induction (**Figure 2—figure supplement 2**). Importantly, though, these results are limited to the 1369 genes interrogated in this study and may or may not reflect how the entire transcriptome of P493-6 cells respond to c-Myc induction.

## Meta-analyses of original and replicated effects

We performed a meta-analysis using a random-effects model to combine each of the effects described above as pre-specified in the confirmatory analysis plan (**Blum et al., 2015**). To provide a standardized measure of the effect, a common effect size was calculated for each effect from the original and replication studies. Cohen's *d* is the standardized difference between two means using the pooled sample standard deviation. The effect size *r* is a standardized measure of the strength and direction of the association between two variables, in this case time during c-Myc induction and gene expression. The estimate of the effect size of one study, as well as the associated uncertainty (i.e. confidence interval), compared to the effect size of the other study provides another approach

**Table 2.** Exploratory statistical tests.

| Genes | Comparison | Study | W value | P value | Sample size (n) |
|---|---|---|---|---|---|
| Active | 0 hr vs 1 hr | RP:CB Lot 1 | 270378 | 0.0103 | 1416 |
| | | RP:CB Lot 2 | 274696 | 0.0395 | 1438 |
| | | *Lin et al. (2012)* | 318799 | 6.67e-05 | 1510 |
| | 0 hr vs 24 hr | RP:CB Lot 1 | 300774 | 7.16e-11 | 1416 |
| | | RP:CB Lot 2 | 324564 | 4.74e-17 | 1438 |
| | | *Lin et al. (2012)* | 400999 | 1.16e-42 | 1510 |
| | 1 hr vs 24 hr | RP:CB Lot 1 | 281679 | 5.45e-05 | 1416 |
| | | RP:CB Lot 2 | 308954 | 1.45e-10 | 1438 |
| | | *Lin et al. (2012)* | 372714 | 4.11e-25 | 1510 |
| Silent | 0 hr vs 1 hr | RP:CB Lot 1 | 187682 | 0.000638 | 1160 |
| | | RP:CB Lot 2 | 174695 | 0.0602 | 1146 |
| | | *Lin et al. (2012)* | 127104 | 0.236 | 1028 |
| | 0 hr vs 24 hr | RP:CB Lot 1 | 185804 | 0.00203 | 1160 |
| | | RP:CB Lot 2 | 184470 | 0.000289 | 1146 |
| | | *Lin et al. (2012)* | 132082 | 0.997 | 1028 |
| | 1 hr vs 24 hr | RP:CB Lot 1 | 166122 | 0.716 | 1160 |
| | | RP:CB Lot 2 | 173608 | 0.0918 | 1146 |
| | | *Lin et al. (2012)* | 136443 | 0.295 | 1028 |

These exploratory statistical tests relate to the data presented in *Figure 2*. Wilcoxon rank sum tests, which treat the data as unpaired, were conducted for the original study (*Lin et al., 2012*) and this replication attempt (RP:CB). Uncorrected *p* values are reported. Sample sizes reported are based on treating genes as unpaired between conditions. Additional details for this experiment can be found at https://osf.io/fn2y4/.

DOI: https://doi.org/10.7554/eLife.30274.008

to compare the original and replication results (*Errington et al., 2014*; *Valentine et al., 2011*). Importantly, the width of the confidence interval for each study is a reflection of not only the confidence level (e.g. 95%), but also variability of the sample (e.g. *SD*) and sample size.

**Table 3.** Exploratory statistical tests.

| Comparison | Study | W value | P value | Sample size (n) |
|---|---|---|---|---|
| 0 hr vs 1 hr | RP:CB Lot 1 | 303897 | 7.78e-50 | 1288 |
| | RP:CB Lot 2 | 278646 | 1.1e-27 | 1292 |
| | *Lin et al. (2012)* | 349351 | 1.27e-130 | 1269 |
| 0 hr vs 24 hr | RP:CB Lot 1 | 272441 | 5.18e-24 | 1288 |
| | RP:CB Lot 2 | 292865 | 7.4e-39 | 1292 |
| | *Lin et al. (2012)* | 368182 | 1.14e-163 | 1269 |
| 1 hr vs 24 hr | RP:CB Lot 1 | 235077 | 7.45e-06 | 1288 |
| | RP:CB Lot 2 | 272028 | 3.73e-23 | 1292 |
| | *Lin et al. (2012)* | 332069 | 5.72e-104 | 1269 |

These exploratory statistical tests relate to the data presented in *Figure 2*. Wilcoxon rank sum tests were conducted for the original study (*Lin et al., 2012*) and this replication attempt (RP:CB) on the difference in expression of active genes during c-Myc induction (e.g. from 0 hr to 24 hr) compared to the difference in expression of silent genes over that same period (e.g. from 0 hr to 24 hr). Uncorrected *p* values are reported. Sample sizes reported are based on number of active and silent genes used in the tests. Additional details for this experiment can be found at https://osf.io/fn2y4/.

DOI: https://doi.org/10.7554/eLife.30274.009

The comparison of total RNA levels at low levels of c-Myc (0 hr) compared to high levels of c-Myc (24 hr) resulted in $d = 4.19$, 95% CI [0.94, 7.37] for the data reported in Figure 3E of the original study (*Lin et al., 2012*). This compares to $d = 0.83$, 95% CI [−0.91, 2.48] for serum lot one and $d = 4.11$, 95% CI [0.90, 7.23] for serum lot two reported in this study. A meta-analysis (*Figure 3A*) of these effects resulted in $d = 2.52$, 95% CI [0.01, 5.03], which was statistically significant ($p=0.0488$). The original and replication results are consistent when considering the direction of the effect, which suggests c-Myc induction increases total RNA levels in P493-6 Burkitt's lymphoma cells. Noticeably, there was substantial within-study variation observed in this replication attempt, due the different serum lots tested. The point estimate of serum lot one was not within the confidence intervals of the original study and serum lot two, and vice versa; however the point estimate of the original study and serum lot two were within the confidence intervals of each other.

There were six comparisons of the gene expression data, three for active genes and three for silent genes (*Figure 3B*). These calculations were performed analyzing the data as paired, for reasons discussed above and as prespecified in the Registered Report (*Blum et al., 2015*). For active genes, expression at 0 hr to 1 hr, 0 hr to 24 hr, and 1 hr to 24 hr the meta-analyses were statistically significant ($p=1.12\times10^{-7}$, $p=7.01\times10^{-4}$, $p=0.0129$, respectively). In all comparisons the results were consistent when considering the direction of the effect; however the effect size point estimate of each study (original, replication serum lot one, replication serum lot two) was not within the confidence interval of the other studies. Further, the large confidence intervals of the meta-analysis along with statistically significant Cochran's Q tests suggest heterogeneity between the original and replication studies. For silent genes, the meta-analysis was not statistically significant for gene expression at 0 hr to 1 hr and 1 hr to 24 hr ($p=0.203$, $p=0.0571$, respectively) and the effect size point estimate of each study was not within the confidence interval of the other studies. Similar to the active gene comparisons, the large confidence intervals of the meta-analysis along with statistically significant Cochran's Q tests suggest heterogeneity between the studies. Furthermore, for the 0 hr to 1 hr comparison the original study and replication studies were in opposite directions, while the 1 hr to 24 hr comparison was consistent. Finally, the comparison between 0 hr and 24 hr for silent genes was consistent when considering direction of the effect with a statistically significant meta-analysis ($p=7.10\times10^{-17}$). The point estimate of the original study was not within the confidence intervals of the replication studies; however both replication studies with different serum lots were within the confidence intervals of the original study and each other. Overall, the gene expression analysis indicates that the effect sizes observed from the two serum lots tested in this replication attempt, although not identical, were more similar to each other than to the original study.

This direct replication provides an opportunity to understand the present evidence of these effects. Any known differences, including reagents and protocol differences, were identified prior to conducting the experimental work and described in the Registered Report (*Blum et al., 2015*). However, this is limited to what was obtainable from the original paper and through communication with the original authors, which means there might be particular features of the original experimental protocol that could be critical, but unidentified. So while some aspects, such as the cell line, induction time course, and the method used to measure gene expression were maintained, others were changed at the time of study design (*Blum et al., 2015*) that could affect results, such as the analytical approach (*Silberzahn et al., 2017*) and serum lot (*Leek et al., 2010*). Furthermore, other aspects were unknown or not easily controlled for. These include variables such as cell line genetic drift (*Hughes et al., 2007*; *Kleensang et al., 2016*) or changes in cellular volume that can impact overall transcript abundance (*Padovan-Merhar et al., 2015*). Whether these or other factors influence the outcomes of this study is open to hypothesizing and further investigation, which is facilitated by direct replications and transparent reporting.

## Materials and methods

As described in the Registered Report (*Blum et al., 2015*), we attempted a replication of the experiments reported in Figures 1B and 3E-F of *Lin et al. (2012)*. A detailed description of all protocols can be found in the Registered Report (*Blum et al., 2015*). Additional detailed experimental notes, data, and analysis are available on the Open Science Framework (OSF) (RRID:SCR_003238) (https:// osf.io/mokeb/; *Lewis et al., 2017*). This includes the R Markdown file (https://osf.io/vdrsh/) that was

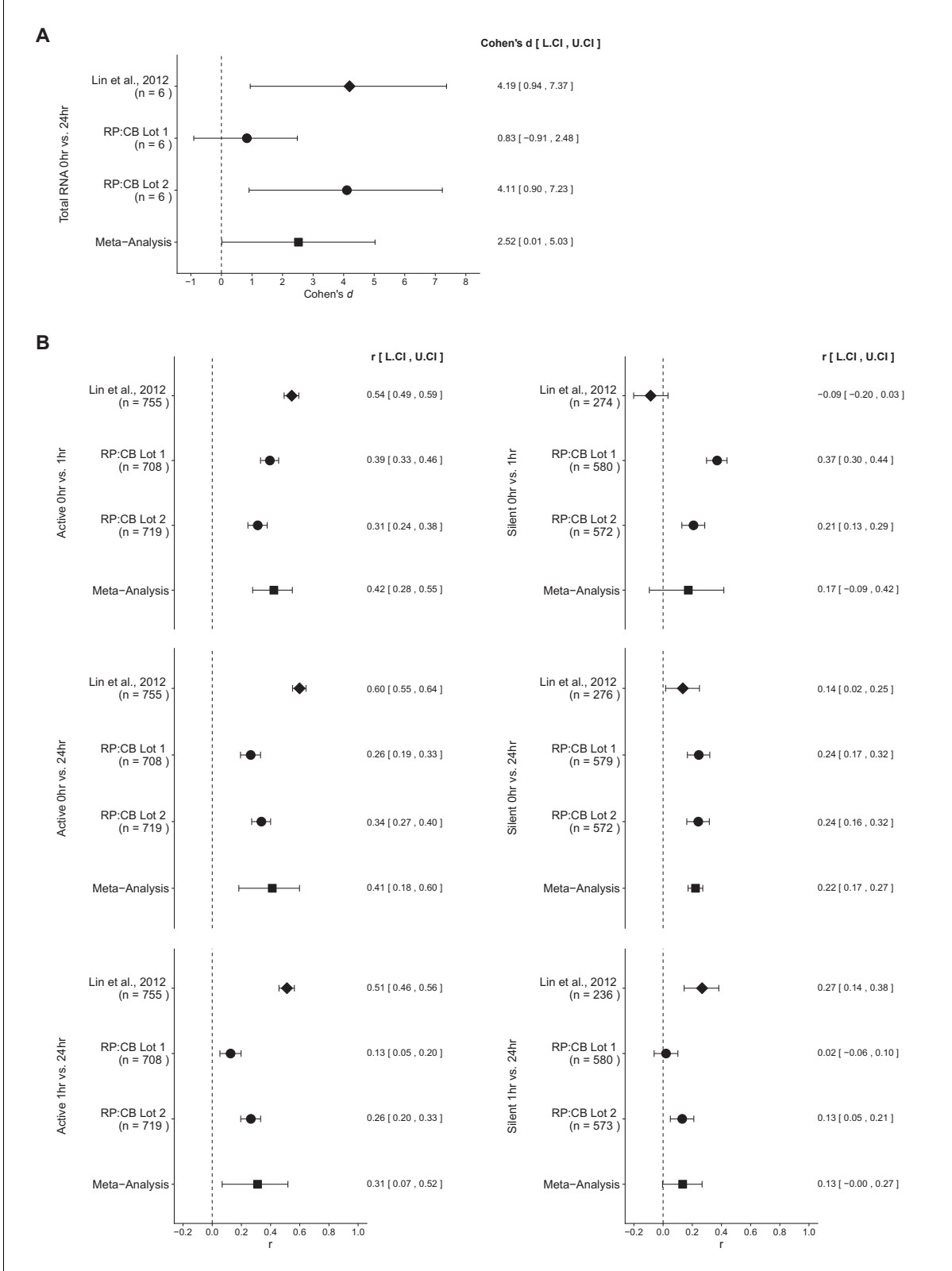

**Figure 3.** Meta-analyses of each effect. Effect size and 95% confidence interval are presented for *Lin et al., 2012*, this replication study (RP:CB), and a random effects meta-analysis of those two effects. Cohen's *d* is the standardized difference between the two measurements, with a larger positive value indicating total RNA levels are increased at 24 hr compared to 0 hr. The effect size *r* is a standardized measure of the correlation (strength and direction) of the association between gene expression and c-Myc induction, with a larger positive value indicating gene expression increased during the

*Figure 3 continued on next page*

*Figure 3 continued*

course of c-Myc induction. Sample sizes used in *Lin et al., 2012* and this replication attempt are reported under the study name. (**A**) Total RNA levels in P493-6 cells 0 hr compared to 24 hr after release from tetracycline (meta-analysis $p = 0.0488$). (**B**) Gene expression of active or silent genes are shown for all comparisons. Active genes: 0 hr compared to 1 hr (meta-analysis $p = 1.12 \times 10^{-7}$), 0 hr compared to 24 hr (meta-analysis $p = 7.01 \times 10^{-4}$), 1 hr compared to 24 hr (meta-analysis $p = 0.0129$). Silent genes: 0 hr compared to 1 hr (meta-analysis $p = 0.203$), 0 hr compared to 24 hr (meta-analysis $p = 7.10 \times 10^{-17}$), 1 hr compared to 24 hr (meta-analysis $p = 0.0571$). Additional details for these meta-analyses can be found at https://osf.io/5yscz/.
DOI: https://doi.org/10.7554/eLife.30274.010

used to compose this manuscript, which is a reproducible document linking the results in the article directly to the data and code that produced them (*Hartgerink, 2017*).

## Cell culture

P493-6 cells (shared by Young lab, Whitehead Institute for Biomedical Research, RRID: CVCL_6783) were maintained in RPMI-1640 supplemented with 1% Ala-Gln and 10% tetracycline-free FBS (Clontech, Mountain View, CA, cat# 631105, lot# 1: A15003, lot# 2: A15032). Cells were grown at 37°C in a humidified atmosphere at 5% $CO_2$. Quality control data for the cell line are available at https://osf.io/e6ftz/. This includes results confirming the cell line was free of mycoplasma contamination (DDC Medical, Fairfield, Ohio). Additionally, STR DNA profiling of the cell line was performed (DDC Medical, Fairfield, Ohio).

For repression of the conditional p*myc*-tet construct in P493-6 cells, 0.1 µg/ml tetracycline (Sigma-Aldrich, St. Louis, MO, cat# T7660) was added to the culture medium and cells were incubated for 72 hr. Under these conditions, P493-6 cells did not proliferate due to a dependency on the expression of *MYC* (*Schuhmacher et al., 1999*). For *MYC* re-induction, cells were washed three times with growth medium and grown in tetracycline-free culture conditions.

## Western blot

P493-6 cells were harvested at the indicated times and total cell lysates were prepared by pelleting ~$1 \times 10^7$ cells (determined with a C-chip disposable hemocytometer) at 4°C at 1,200 rpm for 5 min using a refrigerated centrifuge (Eppendorf, Westbury, NY, model# 5810R). After cell pellets were washed once with ice-cold 1X PBS, pellets were resuspended in RIPA lysis buffer containing 2X SIGMAFAST Protease inhibitors and 2X Phosphatase inhibitor cocktails 2 and 3. Protein concentrations were determined using the Bradford assay according to the manufacturer's instructions. Sample buffer was added to protein lysates and 50 µg of protein along with protein ladder was resolved by SDS-PAGE and transferred to PVDF membrane as described in the Registered Report (*Blum et al., 2015*). The membrane was blocked with 5% w/v nonfat dry milk in 1X TBS with 0.2% Tween-20 (TBST). Membranes were probed with rabbit anti-c-Myc [clone Y69] (Epitomics, Burlingame, CA, cat# 1472–1; RRID:AB_731658); 1:5000 dilution in 5% w/v nonfat dry milk/TBST and mouse anti-ß-actin [clone AC-15] (Sigma-Aldrich, cat# A5441; RRID:AB_476744); 1:10,000 dilution in 5% w/v nonfat dry milk/TBST. Each incubation was followed by washes with TBST and the appropriate secondary antibody: HRP-conjugated donkey anti-rabbit (Sigma-Aldrich, cat# GERPN2124); 1:10,000 dilution in 5% w/v nonfat dry milk/TBST or HRP-conjugated sheep anti-mouse (Sigma-Aldrich, cat# GERPN2124); 1:10,000 dilution in 5% w/v nonfat dry milk/TBST. Membranes were washed with TBST and incubated with ECL Prime Chemiluminescent reagent (Sigma-Aldrich, cat# GERPN2232) according to the manufacturer's instructions. Western blot images were acquired with G:BOX iChem XT and GeneSnap software (RRID:SCR_014249), version 7.12.02 (Syngene, Frederick, Maryland) and quantified using ImageJ software (RRID:SCR_003070), version 1.50i (*Schneider et al., 2012*). All images taken are available at https://osf.io/ujg7t/.

## RNA quantification

P493-6 cells were harvested at the indicated times and total RNA extraction was performed by pelleting ~$1 \times 10^7$ cells (exact number determined with a C-chip disposable hemocytometer) and homogenizing the sample in 1 ml Tri Reagent (Sigma-Aldrich, cat# T9424) according to the manufacturer's instructions. For each sample 10% v/v miRNA Homogenate Additive was added, vortexed, and incubated on ice for 10 min. For each 1 ml of Tri Reagent, 100 µl of bromochloropropane was

added, vortexed for 15–30 s, incubated for 5 min at RT, then centrifuged at 12,000x*g* for 10 min at 4°C. The aqueous phase was recovered and total RNA isolation was performed using the miRVana miRNA extraction kit (Ambion, Waltham, MA, cat# AM1561) according to the manufacturer's instructions. Recovered RNA was eluated in 100 µl nuclease-free water. Total RNA concentrations and purity (data available at https://osf.io/jh5r4/) were measured using a NanoDrop ND-1000 (Thermo Fisher Scientific, Waltham, Massachusetts) with the NanoDrop Operating Software, version 3.3, and converted to ng per 1,000 cells.

## RNA extraction and NanoString nCounter digital gene expression assay

P493-6 cells were harvested at the indicated times and $1 \times 10^6$ cells were collected (number determined with a C-chip disposable hemocytometer) and lysed directly in 100 µl Buffer RLT (Qiagen, Hilden, Germany, cat# 79216) supplemented with ß-mercaptoethanol to yield a concentration of 10,000 cells per µl. This was performed four independent times. Multiple 4 µl aliquots were stored and shipped at −80°C with temperature monitored during shipping to avoid freeze/thaw cycles. Lysates were processed according to the Cell Lysate Protocol (nCounter Gene Expression Assay Manual, NanoString Technologies, Seattle, Washington) according to the manufacturer's instructions and as described in the Registered Report (*Blum et al., 2015*). Three nCounter Reporter CodeSets (nCounter GX Human Immunology Kit, nCounter GX Human Kinase Kit, nCounter Custom CodeSet) encompassing 1369 genes across multiple functional categories were used. Following hybridization, samples were immediately processed with the nCounter Analysis System (NanoString Technologies, NCT-PREP-120). The count data was collected using the nCounter RCC Collector Worksheet (Nano-String Technologies), version 1.6.0 and then positive-, negative-, and housekeeping gene-normalized per nCounter guidelines. Expression for each gene was averaged across the four independent replicate samples. Additionally, for genes that appeared on multiple CodeSets, expression values were averaged together to generate a single value for each gene. A gene was defined as transcriptionally active if its average expression was above one transcript/cell at 0 hr and transcriptionally silent if below 0.5 transcript/cell. A list of all Reporter CodeSets and their expression values (transcripts/cell) are available at *Figure 2—source data 1*. Additional files and analysis scripts are available at https://osf.io/fn2y4/.

## Statistical analysis

Statistical analysis was performed with R software (RRID:SCR_001905), version 3.3.2 (*R Core Team, 2017*). All data, csv files, and analysis scripts are available on the OSF (https://osf.io/mokeb/). Confirmatory statistical analysis was pre-registered (https://osf.io/nj8wb/) before the experimental work began as outlined in the Registered Report (*Blum et al., 2015*). Proposed analysis of gene expression data was conducted by the Wilcoxon signed-rank test using the method proposed by Pratt to handle zero differences (*Pratt, 1959*), with additional exploratory analysis performed using the Wilcoxon rank sum test as reported in the original study and a Wilcoxon rank sum test on the difference in expression of active genes during c-Myc induction (e.g. from 0 hr to 24 hr) compared to the difference in expression of silent genes over that same period (e.g. from 0 hr to 24 hr). Data were checked to ensure assumptions of statistical tests were met. When described in the results, the Bonferroni correction, to account for multiple testings, was applied to the alpha error by dividing the uncorrected value (.05) by the number of tests performed. Although the Bonferroni method is conservative, it was accounted for in the power calculations to ensure sample size was sufficient. In cases where the number of groups were three and the sample sizes were evenly distributed among the groups, Fisher's LSD test was performed resulting in an *a priori* significance threshold of. 05. A meta-analysis of a common original and replication effect size was performed with a random effects model and the *metafor* package (*Viechtbauer, 2010*) (available at: https://osf.io/5yscz/). The sample sizes reported in *Table 1* and *Figure 3* for the gene analysis comparisons is based on the sample size used in the Wilcoxon signed-rank test, which removes samples with zero differences after ranking (*Pratt, 1959*). The raw original study data were shared by the original authors with the summary data published in the Registered Report (*Blum et al., 2015*) and was used in the power calculations to determine the sample size for this study.

## Deviations from registered report

The number of flasks, and thus cells, was increased when tetracycline was added to P493-6 cells to account for the cells not proliferating during this period (i.e. there were two Flask B's as described in the Registered Report, which were pooled prior to seeding). The proposed statistical analysis for the western blot analysis (Protocol 1) described in the Registered Report was not performed since the levels of normalized c-Myc at time 0 hr was at the limit of detection. The number of genes analyzed in the original study, and thus listed in the Registered Report, was reported incorrectly as 1388 instead of 1338 (data shared by original authors). NanoString analysis was conducted using the nCounter RCC Collector Worksheet instead of nSolver Analysis software. Additionally, the statistical tests reported in Figure 3F of the original study incorrectly described the comparisons as between 0 hr and 1 hr instead of between 0 hr and 24 hr (scripts shared by original authors). The corrected values are described above for comparisons and used in the meta-analysis. Additional materials and instrumentation not listed in the Registered Report, but needed during experimentation are also listed.

## Acknowledgements

The Reproducibility Project: Cancer Biology would like to thank the original authors, particular Charles Lin (Baylor College of Medicine) for sharing critical reagents and data, specifically the P493-6 cells. We would also like to thank Courtney Soderberg at the Center for Open Science for assistance with statistical analyses and the following companies for generously donating reagents to the Reproducibility Project: Cancer Biology; American Type and Tissue Collection (ATCC), Applied Biological Materials, BioLegend, Charles River Laboratories, Corning Incorporated, DDC Medical, EMD Millipore, Harlan Laboratories, LI-COR Biosciences, Mirus Bio, Novus Biologicals, Sigma-Aldrich, and System Biosciences (SBI).

## Additional information

### Group author details

**Reproducibility Project: Cancer Biology**
**Elizabeth Iorns**: Science Exchange, Palo Alto, United States; **Rachel Tsui**: Science Exchange, Palo Alto, United States; **Alexandria Denis**: Center for Open Science, Charlottesville, United States; **Nicole Perfito**: Science Exchange, Palo Alto, United States; **Timothy M Errington**: Center for Open Science, Charlottesville, United States

### Competing interests

L Michelle Lewis, Meredith C Edwards, Zachary R Meyers, David Blum: Bioexpression and Fermentation Facility, University of Georgia is a Science Exchange associated lab. C Conover Talbot Jr: Deep Sequencing and Microarray Core Facility, Johns Hopkins University is a Science Exchange associated lab. Reproducibility Project: Cancer Biology: EI, RT, NP: Employed by and hold shares in Science Exchange Inc. The other authors declare that no competing interests exist.

### Funding

| Funder | Author |
| --- | --- |
| Laura and John Arnold Foundation | Reproducibility Project: Cancer Biology |

The funder had no role in study design, data collection and interpretation, or the decision to submit the work for publication.

### Author contributions

L Michelle Lewis, Meredith C Edwards, Zachary R Meyers, David Blum, Acquisition of data, Drafting or revising the article; C Conover Talbot Jr, Analysis, Acquisition of data, Drafting or revising the

article; Haiping Hao, Reproducibility Project: Cancer Biology, Analysis, Drafting or revising the article

### Author ORCIDs
C Conover Talbot Jr (iD) https://orcid.org/0000-0002-3758-2425
Alexandria Denis (iD) http://orcid.org/0000-0002-1210-2309
Timothy M Errington (iD) http://orcid.org/0000-0002-4959-5143

### Decision letter and Author response
Decision letter https://doi.org/10.7554/eLife.30274.015
Author response https://doi.org/10.7554/eLife.30274.016

## Additional files

### Supplementary files
• Source code 1.
DOI: https://doi.org/10.7554/eLife.30274.011

• Transparent reporting form
DOI: https://doi.org/10.7554/eLife.30274.012

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
