## [Decision Letter]

Thank you for submitting your article "Replication Study: Transcriptional amplification in tumor cells with elevated c-Myc" for consideration by *eLife*. Your article has been reviewed by two peer reviewers, and the evaluation has been overseen by Michael Green as the Reviewing Editor and Sean Morrison as the Senior Editor. The two reviewers have opted to remain anonymous.

The reviewers have discussed the reviews with one another and the Reviewing Editor has drafted this decision to help you prepare a revised submission.

Summary:

Lewis et al. performed reproducibility experiments for key data of the paper published by Lin et al. 2012 in Cell. In particular, the experiment described in Figure 1B and its bioinformatic analysis in Figure 3E, F were investigated for reproducibility. Lin et al. studied the change of mRNA levels in response to up-regulated levels of c-Myc in the human B cell line P493-6, which carries a conditional tetracycline-regulated c-Myc gene (Tet-off system). Lin et al. reported that the increase of c-Myc levels in P493-6 cells (after removal of tetracyline) was not sufficient to induce expression of silent genes. In contrast, mRNA levels of those genes, which were expressed already before c-Myc induction, were further elevated. However, the data of Lin et al. (Figure 3F) already showed that this general statement is not valid for all silent genes. Several silent genes (per definition less the 0.5 transcripts/cell) showed elevated mRNA levels (up to 20 transcripts/cell) 24h after c-Myc induction. Importantly, the group of silent genes analyzed in the study of Lin et al. was relatively small (only 514 genes) and randomly selected. If all silent genes in P493-6 cells were analyzed in this study (probably many thousands), the group of silent genes with up-regulated mRNA levels would probably be much larger (hundreds of genes). Thus, the analysis of the data in Figure 3F suffers from the definition of gene sets. The statement of the paper that silent genes are not inducible by c-Myc comes mainly from the fact that the set of analyzed silent genes is very small.

The replication work of Lewis et al. largely confirms the results of Lin et al. (Figure 1B, Figure 3E,F) and shows that elevation of c-Myc levels also elevates the mRNA levels of the same set of genes (potential c-Myc target genes). However, the study of Lewis et al. shows in addition that a larger number of silent genes is inducible by c-Myc and that the rate of induction of c-Myc target genes is generally influenced by the applied serum batch/lot.

Both studies, Lin et al. and Lewis et al., did not address or discuss the question to which extent the increase in mRNA levels is caused by post-transcriptional mechanisms.

For the scientific community two questions are of general (high) interest: How does the entire transcriptome of P493-6 cells respond to increased c-Myc levels and how do all the silent genes respond, neither of which were analyzed in this study?

Essential revisions:

1) The paper of Lin et al. led to a great controversy in the c-Myc field, including the labs of Martin Eilers and Bruno Amati, because it ignored that c-Myc also activates inactive genes and represses active genes. Both labs published 2014 two papers back to back in Nature correcting this biased view of the Lin paper. The biased view of the Lin paper is the result of the definition of small genes sets for the analysis and this issue needs to be explicitly discussed in the reproducibility study.

2) Expression levels of c-Myc are much lower than those of the original study, although the changes of c-Myc expression are in the same direction as the original study (subsection “Total RNA levels following c-Myc overexpression”). Can the authors please discuss this issue?

3) In this study, 580 silent genes were identified with expression level less than 0.5 transcript/cell with a median expression of 0.032. In the original study, 514 genes were identified as silent genes with a median expression of 0.00. If I understand correctly, the current study uses different criteria for a gene to be classified as silent gene. I am curious whether there is a rationale for using 0.5 transcript/cell as the cutoff. In addition, it would be interesting to know how many silent genes are common between the two studies, and how many active genes are common between the two studies (subsection “Digital gene expression following c-Myc overexpression”, first paragraph). The extent of overlap of genes might play a role on some of the inconsistent results between two studies, as mentioned in the meta analysis section (subsection “Meta-analyses of original and replicated effects”, last two paragraphs).

4) The authors should make a R package with a R markdown file and all the associated data.

---

## [Author Response]

Essential revisions:1) The paper of Lin et al. led to a great controversy in the c-Myc field, including the labs of Martin Eilers and Bruno Amati, because it ignored that c-Myc also activates inactive genes and represses active genes. Both labs published 2014 two papers back to back in Nature correcting this biased view of the Lin paper. The biased view of the Lin paper is the result of the definition of small genes sets for the analysis and this issue needs to be explicitly discussed in the reproducibility study.

We agree that the small gene set analyzed in the original study and this replication attempt is more biased than a comprehensive analysis of all genes. We also agree that dichotomizing gene expression (which is a continuous variable) into active and silent genes has additional negative consequences. We have expanded the manuscript to discuss these issues and are including additional supplemental figures that present the gene expression results on a continuous scale, instead of only segmented into silent and active genes. Thus, while this replication study is prone to the small gene set, due to replicating the design of the original study, the additional figures provide additional means to assess the impact c-Myc induction has on gene expression.

2) Expression levels of c-Myc are much lower than those of the original study, although the changes of c-Myc expression are in the same direction as the original study (subsection “Total RNA levels following c-Myc overexpression”). Can the authors please discuss this issue?

Thank you for raising this question. We have expanded the section highlighted to discuss some potential factors that might account for this difference.

3) In this study, 580 silent genes were identified with expression level less than 0.5 transcript/cell with a median expression of 0.032. In the original study, 514 genes were identified as silent genes with a median expression of 0.00. If I understand correctly, the current study uses different criteria for a gene to be classified as silent gene. I am curious whether there is a rationale for using 0.5 transcript/cell as the cutoff. In addition, it would be interesting to know how many silent genes are common between the two studies, and how many active genes are common between the two studies (subsection “Digital gene expression following c-Myc overexpression”, first paragraph). The extent of overlap of genes might play a role on some of the inconsistent results between two studies, as mentioned in the meta analysis section (subsection “Meta-analyses of original and replicated effects”, last two paragraphs).

Both studies used the same criteria to classify a gene as silent (less than 0.5 transcript/cell) and active (greater than 1 transcript/cell) at time 0 hr. The reason for the different number of genes, as well as the median expression at time 0 hr, is because the original study reported a majority of the silent genes as having an expression of 0.00 transcript/cell. We have revised the text to clearly state that the same criteria were used. We are not aware of what the rationale is for using these cutoffs, as it was not included in the original study. As stated above in response to comment 1, there are also negative consequences of dichotomizing continuous variables. To provide another means of displaying the data, we included additional supplement figures to illustrate the distribution of gene expression of all genes analyzed at the different times (Figure 2—figure supplement 2).

We agree that the extent of overlap of genes between the two studies (and the two lots tested in this replication attempt) are of interest. We analyzed what percent of genes were common between the different studies for active and silent genes and have included this in the revised manuscript. There was 88.8% commonality for active genes between serum lot one in this replication attempt and the original study (90.1% for serum lot two) with 96.4% commonality for silent genes between serum lot one in this replication attempt and the original study (95.7% for serum lot two).

4) The authors should make a R package with a R markdown file and all the associated data.

We agree that sharing the Rmd file and associated scripts and data are important. We used the OSF to create a project that stores all files/scripts/methods associated with this study and have a function so the associated data and scripts are downloaded when knitting the Rmd file. This will work now while the project is private as well as when the project, with all the associated files, is made public. The location of the Rmd file is also included in the Materials and methods section of the manuscript (https://osf.io/vdrsh/) and we’ve included it as a ‘Source code file’ during resubmission.